

**Coseismic deformation field derived from Sentinel-1A data and slip inversion of**
**the 2015 Chile Mw8.3 earthquake**
Zuo Ronghu, Qu Chunyan, Shan XinJian, Zhang Yingfeng, Zhang Guohong, Song Xiaogang,
Liu Yunhua, Zhang Guifang
State Key Laboratory of Earthquake Dynamics, Institute of Geology, China Earthquake Administration, Beijing,
100029, China
*Correspondence to:* Qu Chunyan (dqyquchy@163.com)

9       **Abstract.** We obtain the coseismic surface deformation fields caused by the Chile Mw8.3 earthquake on 16

September 2015 through analyzing Sentinel-1A/IW InSAR data from ascending and descending tracks. The results
show that the main deformation field looks like a half circle convex to east with maximum coseismic displacement
of about 1.33m in descending LOS direction, 1.32m in ascending LOS direction. Based on an elastic dislocation
model in a homogeneous elastic half space, we construct a small-dip single plane fault model and invert the coseismic
fault slip using ascending and descending Sentinel-1A/IW data separately and jointly. The results show that the
patterns of the main slip region are similar in all datasets, but the scale of slip from ascending inversion is relatively
smaller. Joint inversion can display comprehensive fault slip. The seismic moment magnitude from the joint
inversion is Mw8.25, the rupture length along strike is about 340 km with a maximum slip of 8.16m near the trench
located at -31.04N, -72.49E, and the coseismic slip mainly concentrates at shallow depth above the hypocenter with
a symmetry shape. The depth where coseismic slip is near zero appears to a depth of 50km, quantitatively indicating
the down-dip limit of the seismogenic zone. From the calculated coseismic Coulomb stress change, we find
aftershocks locations correlate well with the areas having increased Coulomb stress and most areas with increased
Coulomb stress appeared beneath the main shock fault plane.
**Keywords:** Chile earthquake;InSAR;Coseismic deformation; slip distribution; Coulomb stress change

**1.  Introduction**

On September 16, 2015, a magnitude 8.3 offshore earthquake struck west of Illapel, Chile. About one million

people evacuated from their homes. A tsunami hit Coquimbo and other villages with waves close to 4 meter high
(http://www.emsc-csem.org/Earthquake/237/M8-3-OFFSHORE-COQUIMBO-CHILE-on-September-16th-2015-
at-22-54-UTC). This huge earthquake was the result from thrust faulting on the interface between the Nazca and
South America plates in central Chile,of which the epicenter is about 85 km to the Chile Trench. At the latitude of
this event, the Nazca plate is moving towards the east-northeast at a velocity of 65-74 mm/yr with respect to South
America, and begins its subduction beneath the continental South American plate at the Peru-Chile Trench. Located
on the subduction boundary of the two plates, Chile is one of the most earthquake-prone countries in the world
(Madariaga et al., 2010). The 2010 Mw8.8 Maule earthquake in central Chile ruptured a ~600 km (Tong et al, 2010;
Delouis et al, 2010; Pollitz et al, 2011) long section of the plate boundary south of this 2015 event and the 1985
Mw7.8 event (Barrientos,1995). This subduction zone hosted the largest earthquake on the record, the 1960
magnitude 9.5 Chile earthquake (National Earthquake Information Center (NEIC), 2010; Cifuentes et al. 1989). In
the past century, the region within 400 km to the event on September 16, 2015 has suffered 15 other earthquakes
with magnitude greater than 7 (http://earthquake.usgs.gov/earthquakes/eventpage/us20003k7a#general_summary).
Along the trench, most zones has been ruptured during the past earthquakes (Vigny et al., 2011) (Fig. 1).


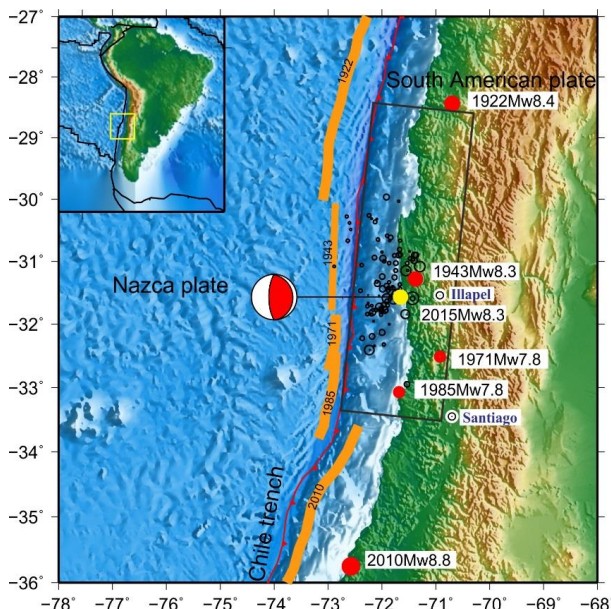

**Figure 1.** Tectonic setting around the 2015 Mw8.3 earthquake. Red dots: Epicenters of historical earthquakes near this Illapel 2015 event (Mw8.4 in 1922, Mw8.3 in 1943, Mw7.8 in 1971, Mw7.8 in 1985, Mw8.8 in 2010). Yellow dot: Epicenter of 2015 Mw8.3 Illapel event. Black circles: aftershocks (from http://earthquake.usgs.gov, as of 18/9/2015). The red barbed line is the Chile trench trace. Color stripes along the trench depict past earthquake rupture zones (adapted from Vigny et al., 2011). ETOPO1 Digital Elevation Models (http://www.ngdc.noaa.gov/mgg/global/global.html) were used to generate the background topography. The black rectangle is the fault plane projected onto the surface.

The information on the down-dip limit of the seismogenic zone and transition depth from seismic to aseismic slip of thrust faulting earthquakes is important to understand Chile subduction zone. (Mendoza et al.1994; Pritchard et al. 2006). Modern geodetic technology can obtain small deformation of crust and could be used as a tool for seismic hazard assessment (Ader et al. 2012). Inversion of the co-earthquake rupture depth constrained by a dense geodetic data, e.g., deformation measurement from Synthetic Aperture Radar Interferometry (InSAR) and Global Positioning System, permits to address this issue. For example, using InSAR and GPS data, Tong et al. (2010) estimated the maximum rupturing depth of the 2010 Chile Mw8.8 event, which is 43-48km and is largely consistent along the 600km-long rupture zone. For the same event and also using joint inversion of ALSO/PALSAR and GPS data, Pollitz et al. (2011) suggested that the fault rupture of this event terminated at a depth of 35km, which is relative shallow, and likely associated with the spherical layering Earth model used in their inversion. Using joint inversion of teleseismic records, InSAR and high rate GPS (HRGPS) data, Delouis et al. (2010) constrained the maximum down-dip depth as 50km for the 2010 Chile great shock. These studies on the rupturing depth of the great earthquakes can provide evidence for determining the seismogenic depth, lower limit of stick-slip and the boundary between seismic and aseismic layers in the subduction zone beneath central Chile.

In this work, both the descending and ascending track of Sentinel-1A/IW data have been downloaded and processed to reconstruct the coseismic deformation field of the 2015 Chile event. Then, we inverted the slip distribution on the seismogenic fault plane of this earthquake with three different constraints of descending and ascending measurements separately and jointly. Thirdly, we discussed the observation and inversion results.



Additionally, inversion results (e.g., rupture depth) from each track (ascending, descending) and join tracks has been
analyzed. Finally, in order to identify the promoting relationship between the main shock and aftershocks, we
estimated the shear stress in the aftershock area of the main shock event. The complete InSAR coverage over the
rupture area provided a unique information to derive a detailed slip model, which is needed to estimate the spatially
varied stress change from the event.
**2. Sentinel-1A InSAR data and processing**
We investigated the crustal deformation triggered by the 2015 Mw8.3 Chile earthquake using interferometric
synthetic aperture radar (InSAR) with Sentinel-1A Interferometric Wide Swath (/IW) mode data in both descending
and ascending orbits. Sentinel-1A satellite was launched by ESA on April 3, 2014, and its IW Mode used the
advanced TOPSAR (Terrain Observation with Progressive Scans SAR) technology
(https://sentinel.esa.int/web/sentinel/user-guides/sentinel-1-sar/applications). The radar image in IW model has a
swath width up to 250km, spatial resolution of 5m×20m (single look), and revisit period of 12 days, providing a
good data source for large-scale monitoring of ground deformation. At present, Sentinel-1A satellite data can be
accessed through ESA data hub (https://scihub.esa.int/). It is one and three days after the 16 September 2015 Chile
event, i.e. 17 September and 19 September that Sentinel-1A acquired descending and ascending data covering the
coseismic area. The selected post-earthquake images are close to the event time, whereas the pre-image can be
acquired long before the event. Because data very close to the mainshock time, permit to study the coseismic
deformation of this event without much aftershock deformation. As the affected area of this great event is very large,
we use three adjacent frame along the same descending track to get a full coseismic deformation field. Since, in the
ascending track, only two frames are available, we get only part of coseismic deformation field. The SAR data and
its parameters used in this paper are shown in Table 1.

**Table 1.** The Sentinel-1A/IW data used in this study

| Number | Track | Master | Slave | Average Perpendicular Baseline(m) | Average Ambiguity Height(m) |
|---|---|---|---|---|---|
| 1 | Descending | 20150707 | 20150917(north) | 1 | 13667 |
| 2 | Descending | 20150707 | 20150917(middle) | -1 | 13667 |
| 3 | Descending | 20150707 | 20150917(south) | -3 | 4556 |
| 4 | Ascending | 20150826 | 20150919(north) | 73 | 187 |
| 5 | Ascending | 20150826 | 20150919(south) | 70 | 195 |

We used the GAMMA software to process the Sentinel-1A data. The interferograms have been processed
separately for each frame along the same track, and then mosaicked to a signal wrapped differential interferogram.
To reduce noise, multi-look processing of 10-sight in range and 2-sight in azimuth directions were performed to the
interferograms. It requires a very high SLC (single look complex image) registration accuracy in azimuth direction
(Meta et al., 2010). To achieve an accuracy of a very small fraction of an SLC pixel nearly 0.1/% in azimuth direction,
we performed intensity image based and iterating offset estimation for many times until the azimuth offset correction
became at least smaller than 0.02 SLC pixel. Meanwhile, the adaptive filters based on interferometric fringe
frequency and gradually decreasing windows were applied to interferograms so that their ratio of signal to noise was
highly enhanced, fringes associated with seismic deformation were highlighted. The algorithm of minimum cost
flow (Werner et al., 2002) was implemented for phase unwrapping with Delaunay triangle network that is suitable





for low coherent areas. To make phase continuous and smooth, before integration of mosaic three adjacent
interferograms on descending track, we firstly unwrapped the interferogram in the southernmost of the study area,
and used the far field to its south as the start point for unwrapping. It was followed by unwrapping the interferogram
in the middle, using the same-place point in superposed portion of the two adjacent interferograms as the reference
and initial phase value for unwrapping it. Similarly, the interferogram in the north was unwrapped. Consequently,
the interferogram from integrating these three images was featured by continuous phase without signature of
boundaries. When doing this, we removed topographic phase by the generating simulated interferogram using
ASTER GDEM data (30m×30m) (http://gdem.ersdac.jspacesystems.or.jp/search.jsp).

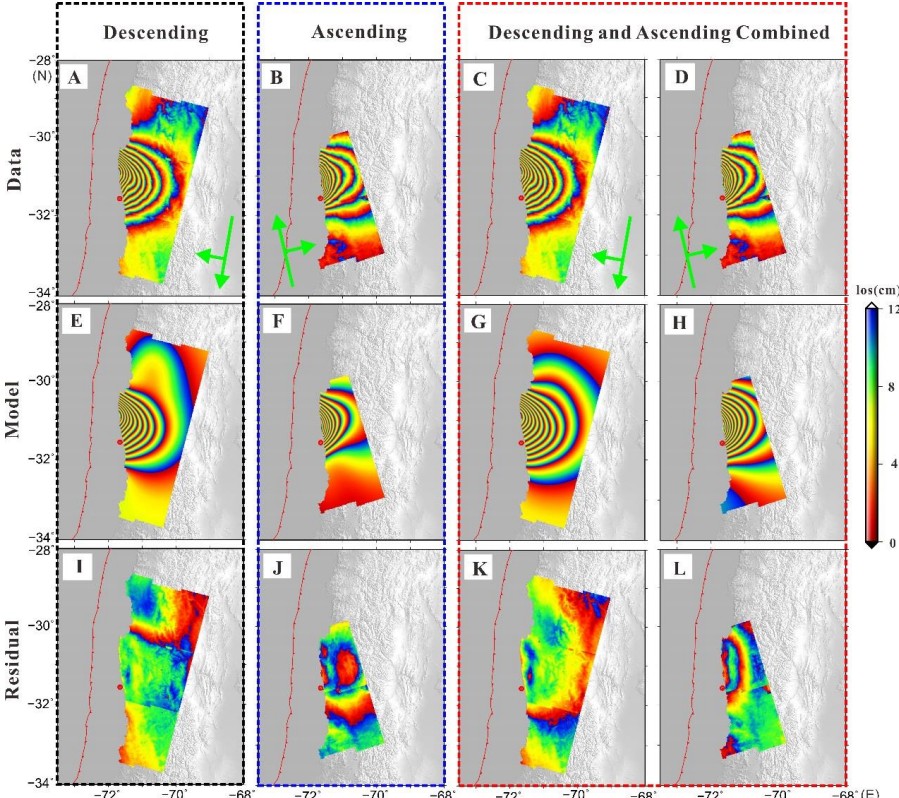


**Figure 2.** Earthquake mainshock, as seen from radar satellites that allow quantifying displacements in the Line of

Sight (LOS) as indicated by green arrow in (**A-D**). Upper row is the data (**A**, **C**, coseismic deformation field from

descending track; **B**, **D**, coseismic deformation field from ascending track). Center row (**E-H**) is the model solution

(the model in black dashed area is constrained by descending data, in blue dashed area is constrained by ascending

data, in red dashed line is constrained by ascending and descending data combine). Bottom row (**I-L**) shows the

residual after subtracting the model from the data. Red dot depicts the location of the mainshock epicenter from

USGS, about 10km to the coast. The red barbed line is the trench trace.


**3.   Coseismic deformation fields derived from Sentinel-1A ascending and descending InSAR data**

As mentioned above, the deformation field from descending data was generated by integrating three

interferograms along the same track, which covers almost the whole affected area of the 2015 Chile earthquake (Fig.
2A). Towards the continent, the fringes become progressively sparse, implying decreasing gradients of deformation.





While we set the deformation in far field, without any phase change, to zero, the maximum LOS displacement is -
133cm near the coast. It looks like a half circle convex to east, with most of data being negative, which means
subsidence in descending LOS direction. According to the full descending track fringes, the deformation area is
within 300km long in the NS direction or along the coast, and 190km in the EW direction. Although the deformation
field derived from the ascending data is only based on two frame, it also covers the major part of the seismic
deformation area (Fig 2B), consistent with that from the descending data. This field is also of a half circle convex to
east with maximum LOS displacement 132cm (far field deformation be zero). Deformation in LOS being positive,
means uplift in ascending LOS direction. The positive and negative with similar magnitude of LOS deformation
from ascending and descending data suggest that the crustal deformation caused by this earthquake is dominated by
horizontal motion.

**4.    Fault slip inversion and interferogram simulation**
**4.1  Inversion method and fault model construction**
Focal mechanism solutions given by USGS (http://www.usgs.gov/) and GCMT
(http://www.globalcmt.org/CMTsearch.html) show that the seismogenic fault of this earthquake is thrust with a small
dip angle. Its surface trace closely follows the trench axis. Based on the focal mechanism solutions, aftershock
distribution and InSAR deformation fields obtained in this work, we built a single-plane fault model in elastic half-
space (Okada Y. 1985) to invert the static coseismic slip distribution on the rupture surface constrained by the
Sentinel-1A descending and ascending data both separately and jointly. The linear-inversion, Sensitivity Based
Iterative Fitting (SBIF) method (Wang et al., 2008) was employed. Firstly, the fault plane was divided into multiple
fault patches. Each patch was presumed to slip uniformly. In this way, the non-linear problem can be transformed
into a linear problem. Then we used the mean square deviation reducing function to quantify the misfit between the
simulated interferogram and the observed one. Using this function, by minimizing mean square deviation, non-
uniform slip distribution on the fault plane can be determined. The mathematical formulation of the inversion is
expressed by:

$$f(s) = \sum_{k=0}^{K} \left\| D_k - D_k^0 - G_k s(x) \right\|^2 + \beta^2 \left\| Hs \right\|^2 \to \min$$


Where 's(x)' is slip vector, 'k' is the patch index of different input data set. 'D' is the matrix of observation data, 'D⁰'
is the static offset of the observations, 'G' is the Green's function for an elastic half-space, which describes the
relation between the model prediction and the observation. 'β' is defined as the smoothing factor, 'H' is the Laplacian
operator, and $\left\| Hs \right\|^2$ represents the slip roughness. Assuming a Poisson ratio of 0.25 and using SBIF program, we
calculate the Green's functions of the homogeneous elastic half-space model using Okada.
Also, we resampled the InSAR deformation field by the quad-tree resampling method for inversion (Jónsson
et al., 1999; Lohman et al., 2005). The reason to do quad-tree is to reduce computation load and also to keep the
pattern of deformation map. In the resampling process, we have 12763 sampled points from Sentinel-1A descending
data and 9196 sampled points from Sentinel-1A ascending data, respectively, which still have a much higher spatial
density than other geodetic data (e.g., GPS). The initial fault geometry is a single planar surface striking N4.6°E and
dipping toward the east, where it takes trial values between 10° and 30°. The rake angles are in the range of 80°~150°.
The upper boundary of the fault is to surface. The initial fault model is steadily modified through optimal fitting to
the deformation fields derived from both Sentinel-1A descending and ascending data sets in joint inversion. The
final fault model is that dip angle is 18.3°,strike is 4.6°,and the fault dimensions are 535 km along-strike and
200 km down-dip, with fault plane divided into many rectangular patches whose grid is 10km×10km from best fault
resolution test (Table 2).





**Table 2.** Main parameters of the optimal fault model

| Parameters | Lat_Ref(°) | Lon_Ref(°) | Strike(°) | Dip(°) | Length(km) | Rake(°) | Width(km) | Top_Depth(km) |
|---|---|---|---|---|---|---|---|---|
| Final values | -33.3 | -72.75 | 4.6 | 18.3 | 535 | 80~150 | 200 | 0 |


**4.2 Fault slip inverted from Sentinel-1A descending and ascending data separately and jointly**
Using the fault model and resampled data points described above, we inverted slip distribution on the fault
plane of this Chile earthquake constrained by the Sentinel-1A descending and ascending observations separately and
jointly. The calculated residual maps between the observed and simulated ones are shown in Figure 2. The result
shows that when the inversion is constrained by Sentinel-1A descending data alone, the preferred slip model shows
a preponderant fault rupture zone located in the shallow part of the up-dipping thrust fault above the hypocenter (Fig.
3B). The maximum fault slip is over 8 m at a shallow depth, located in the northwest of the epicenter. The down-dip
boundary of the rupture zone is relatively clear, and its depth is only about 35km under the surface. The rupture
length of the slip area is about to 340km, comparable to 335 km of the major axis of aftershock distribution in north
and south direction. But the main slip is concentrated in a shallow region that is 15km deep and 200 km long on the
subduction interface. The mean rake angle from inversion is 110°, consistent with the thrust fault motion. The
simulated interferogram is reconciled well with the observed one, with fitting degree 99.99%. The seismic moment
magnitude is Mw8.27.
When the inversion is constrained by the Sentinal-1A ascending data alone, the resulted fault slip magnitude
and its scope are all smaller than that constrained by the Sentinel-1A descending data, although the overall patterns
of the slip region in both cases are similar (Fig. 3A). The probable reason is that the ascending data do not fully
cover the deformation field, since only two frame images are available. The maximum slip from this inversion is
only about 3.43m. The mean rake angle is 102.42°. The simulated interferogram fits the observed one very well with
the fitting degree 99.97%. The seismic moment magnitude is Mw8.09. It should be pointed out that the down-dip
boundary of the rupture zone is much deeper, with the depth about 50km under the surface.

**Table 3.** Fault plane and source parameters of the 2015 Chile earthquake given by teleseismic focal mechanism
solutions and this study

| Source | Latitude (°) | Longitude (°) | Depth (km) | Mw | Strike (°) | Dip (°) | Rake (°) | Scalar Moment(N.m) |
|---|---|---|---|---|---|---|---|---|
| GCMT | -31.22 | -72.27 | 17.8 | 8.2 | 5 | 22 | 106 | $2.455\times10^{21}$ |
| USGS | -31.57 | -71.67 | 20.7 | 8.3 | 5 | 22 | 106 | $3.467\times10^{21}$ |
| Descending | - | - | - | 8.27 | 4.6 | 18.3 | 116.41 | $3.126\times10^{21}$ |
| Ascending | - | - | - | 8.09 | 4.6 | 18.3 | 102.42 | $1.679\times10^{21}$ |
| Jointly | - | - | - | 8.25 | 4.6 | 18.3 | 103.24 | $2.917\times10^{21}$ |


We implemented a fault slip inversion jointly using the Sentinel-1A ascending and descending data with equal
weight. The result falls between the two inversion results by using the two data alone (Fig. 3C). The shape of slip
area seems to be symmetrical. The inversion result indicates that the mean rake angle is about 103.24°, which is
in agreement with a thrust fault. The fitting degree is also very good, about 99.97%. The maximum slip is about
8.16m at shallow depth near the trench. The seismic moment magnitude is Mw8.25. The final inversion results are
shown in Table 3 and Figure 3. It seems that combination of descending and ascending InSAR data used in inversion
helps to derive a more comprehensive fault slip distribution.




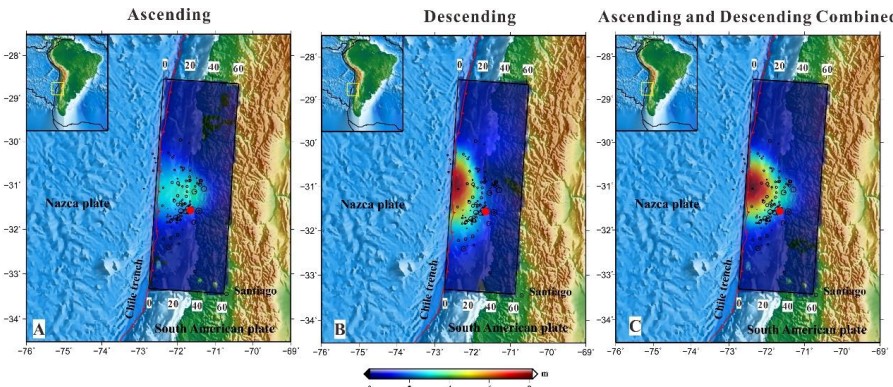

**Figure 3.** Fault slip distribution inverted by using Sentinel-1A ascending and descending data. The fault trace at the surface is from (-33.3N, -72.75E) in south to (-28.5N, -72.30E) in north, strike is N4.6°E. The blue rectangle is the fault plane projected onto the surface. Number with white background is the depth of the fault in kilometer. Red dot is the position of epicenter from USGS. Aftershocks (from http://earthquake.usgs.gov, as of 18/9/2015) are represented by black circle. **(A)** Fault slip distribution inverted by ascending data. **(B)** Fault slip distribution inverted by descending data. **(C)** Fault slip distribution inverted by ascending and descending data jointly as constraints.

### 4.3 Static Coulomb stress changes

In order to identify the promoting relationship between the main shock and aftershocks, we calculated the coseismic Coulomb Failure Stress (CFS) change on the fault plane and surrounding medium by using the optimal slip model (Lin 2004; Toda 2005), which comes from inversion by ascending and descending jointly. Computing the CFS change following an earthquake tells whether a fault has been brought closer or away from rupture (Stain 1999). Many researches suggest that 0.1bar shear stress change can have a great influence in earthquake activities (King et al., 1994). From the location of aftershocks, we found most of the aftershocks happened under our inverted fault plane (Fig. 4B red line). The distribution of aftershocks reflected a special plane (Fig. 4B dash line in blue) whose dip angle is about 31°, much bigger than the dip angle (18.3°) from our main shock inversion and results from USGS (dip=19°) and GCMT (dip=22°). So in our models we set the receiver plane dip angle 31°, with strike 4.6° and rake 105°, to see what the static coulomb stress the main shock promoted to aftershock is. From the coseismic Coulomb stress profile at 30km depth (Fig. 4A), we estimate the coseismic shear stress change ranged from -12 bar (stress drop) to 8 bar (stress increase) and find aftershocks (depth in 20km-30km) locations correlate well with the areas having increased Coulomb stress. The three special profiles in vertical (dip=90°) reflected most areas with increased Coulomb stress appeared beneath the main shock fault plane, which is consistent with the location where aftershocks took place. At the same time, we can see static Coulomb stress up the main shock fault plane is released (Fig. 4C). A frictional ratio of 0.4 and rake angle 105 were used in these results, but we also explored different frictional ratios (0.3-0.7) and rake angles of receiver plane (100°-110°). No significant difference was observed for the obtained CFS, implying that the models are robust.


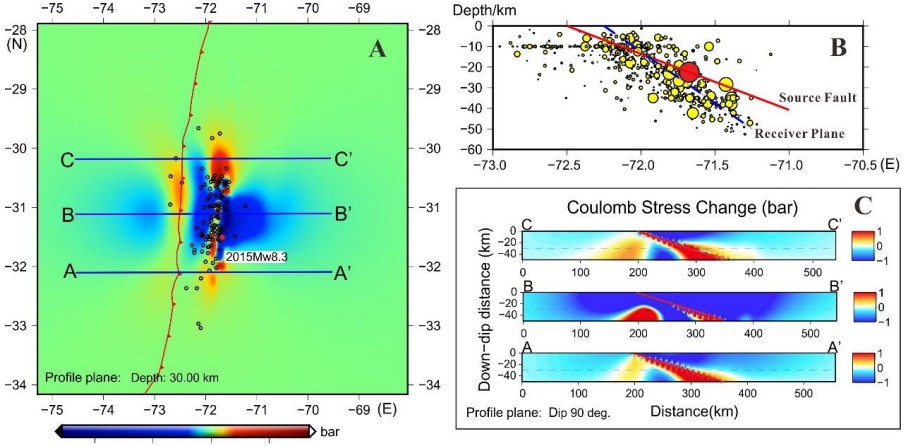

**Figure 4. (A)** Coulomb Failure Stress (CFS) changes (in bar) due to 2015 September 16 main shock is calculated at a depth of 30km. The red circle marks the main shock and black circle is the location of other aftershocks (Mw>4) in the depth from 20km to 30km. **(B)** Location of aftershocks (from http://earthquake.usgs.gov, as of 11/10/2015) in vertical plane. The main shock is the red dot and the red line is the fault plane in our inversion whose dip angle is 18.3 °. Blue dash line is the receiver plane which is best-fit from the location of aftershocks. The dip angle of the receiver plane is about 31 °. **(C)** Cross section with the main fault lines (red line) and CFS calculated by the coseismic mainshock slip model, where red denotes a stress built up, and blue a stress shadow. The areas beneath the fault plane receive large CFS built up.

## 5. Discussion

This paper presents a study of 2015 Chile Mw8.3 earthquake based on Sentinel-1A InSAR data of ascending and descending tracks. The purpose is to investigate the coseismic deformation and invert the slip distribution on its fault plane and rupture depth. The results show that the overall slip area is located in the shallow portion of the subduction interface between the source and the trench, with a NS symmetric pattern. The moment magnitude (8.25) and seismic moment ($2.917 \times 10^{21}$ Nm) are between the results of GCMT (Mw=8.2,$M_0$=$2.455 \times 10^{21}$) and USGS (Mw=8.3,$M_0$=$3.467 \times 10^{21}$), consistent with focal mechanism solutions from seismic waves. It indicates that the inversion results of this work are reliable, including the dip angle 18.3 °which is consistent with result from USGS (dip=19 °) and GCMT (dip=22 °). Fault slip is likely related with the ground broader deformation field in EW direction from our work. To assess the resolution capabilities and stability of the fault model, we conducted fault resolution tests of slip identification and find sub-fault grid 10km×10km is best.

The coseismic deformation fields of the Chile event derived from Sentinel-1A descending and ascending data are roughly consistent in the shape. The positive and negative with similar magnitude of LOS deformations from ascending and descending data suggest that the crustal deformation caused by this earthquake is dominated by horizontal motion. The fault-slip distributions of the Chile event from inversions constrained by different data sets (Sentinel-1A descending, Sentinel-1A ascending, and jointly) have both similarities and disparities. The common points include that the mean rake angles (102 °-106 °), indicating a thrust fault with slight right lateral slip,and the outlines of slip regions are about in the same area from the three inversions. There are differences between the estimated slip (~3m) from inversion using Sentinel-1A ascending data alone and slip (~8m) from inversion using Sentinel-1A descending data alone or inversion jointly. An alternative explaination is that the ascending displacement field is smaller than the actual one because two-frame images cannot cover the whole coseismic deformation field.



With reference to all the results, we suggest that the slip-distribution from the inversion using Sentinel-1A descending
and ascending data jointly seems to be more convincing.
Here we also compare the surface deformation fields and slip distributions on the fault planes of the 2015
Mw8.3 and 2010 Mw8.8 Chile events. Although the South American subduction zone hosts a significant number of
large earthquakes,only these two events have InSAR data available for such a comparison. InSAR data,owing to
the advantages of dense sampling, can provide the best constraint on the slip location, distribution and depth on the
rupture plane by quantitative measuring the static displacement on the ground surface caused by an earthquake. We
find that the two events are different in coseismic deformation. For the 2010 event, the deformation spreads along
the coast with at least two centers (Tong et al., 2010; Bertrand et al, 2010). The slip distribution from inversion is
also of a narrow long strip, rupturing over 600km. The slip is concentrated on the north and south of the source,
mostly at depths of 15km-25km, and no large slip at the trench. In contrast, the deformation field of the 2015 event
is a complete half circle shape. The inverted slip concentration area is nearly NS symmetric, close to the shallow
trench. The 2015 event is located over 400km north of the 2010 shock, both on the subduction slab of the Nazca
plate beneath the South American plate. Both events are interplate thrusts with similar tectonic and dynamic settings.
But as mentioned above their rupture features are different. The analysis suggests that the 2015 event has a shallow
source (25km) and a connective rupture in up-dip direction above the source reaching the trench. Meanwhile its main
shock occurred on a big barrier. While the 2010 event is relatively deeper (33km), and at least ruptured two big
barriers. It may induce to speculate that the subduction zone has many barriers of varied sizes on different segments.
And the coupling or locking degrees are variable at different sections of the subduction zone.
However, the maximum rupture depth (50km) of the 2015 Mw8.3 event from the model of this work is roughly
consistent with the rupture depth of the 2010 Mw8.8 shock derived from inversion of previous studies, which are
based on InSAR plus GPS or InSAR, GPS, and seismic wave data (Tong et al., 2010; Bertrand et al, 2010). It is also
in accordance with the depths of the subduction zone in northern Chile (Tichelaar et al., 1993; Delouis et al. 1997)
and sourthern Chile (Delouis et al. 2009) and the locking depth of this zone from GPS data (Ruergg et al., 2009).
Previous studies suggest that this depth is the transition between the seismic and aseismic layer in the subduction
zone beneath the South American plate. This work further confirms this conclusion.

## 6 Conclusions

In this work, we obtained the coseismic deformation field of the 2015 Chile Mw8.3 earthquake using Sentinel-
1A descending and ascending data. The positive and negative with similar magnitude of LOS deformation from
ascending and descending data suggest that the crustal deformation is dominated by horizontal motion. The inversion
constrained by Sentinel-1A ascending and descending data jointly can display comprehensive fault slip. We find the
strike angle N4.6°E, the fault dimension 535km (along-strike) × 200km (down-dip), and the dip angle 18.3 can fit
our model better. Mean rake angle from inversion is 103.24°, which indicates a thrust fault with slight right-lateral
slip. A maximum slip of 8.16m on the fault plane appears near the trench, the length of rupture reaches about 340km
along strike but mainly extending to north side of the epicenter, and the overall slip pattern is moderately symmetrical,
with the down-dip end of the rupture at about 50km which is roughly consistent with the rupture depth of the 2010
Mw8.8 shock. The seismic moment magnitude is Mw8.25, the scalar moment from jointly inversion is
$(2.917 \times 10^{21} \text{N m})$, and the fitting degree of the whole field is 99.97%. Coseismic Coulomb stress change reflected
most areas with increased Coulomb stress appeared beneath the main shock fault plane, which is consistent with the
location where aftershocks took place. At the same time, we can see static Coulomb stress above the main shock
fault plane is released.





**Acknowledgements**: We would like to thank ESA for providing free Sentinel-1A data, we thank Wang Rongjiang
for providing SBIF program. This work was supported by the National Natural Science Foundation of China
(41374015, 41411011073) and National Key Laboratory of Earthquake Dynamics (LED2015A03, LED2013A02).
All figures were generated with Generic Mapping Tools software.

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
