# Peer review of "Coseismic deformation field derived from Sentinel-1A data and slip inversion of"

_Natural Hazards and Earth System Sciences, 2015_

## Referee Comment (RC1) · Anonymous Referee #1 · 18 Mar 2016

The paper "Coseismic deformation field derived from Sentinel-1A data and slip inversion of the 2015 Chile Mw8.3 earthquake" present surface deformation associated with the past year Chilean earthquake evaluated using the new ESA satellite Sentinel 1-A in wide swath mode. The data are then modeled with a very simple (probably oversimplified) model using an elastic half space and simulating the fault plane as a single flat surface. The fault slip computed by this inversion is then used to compute Coulomb failure stress and compared it to the aftershock distribution. The paper, in particular the last two part of it is very problematic from a scientific point of view. The English of the full paper need major reworking, with presence of many colloquialisms (eg. line 29 "huge" earthquake), sentences that do not make any sense (e.g. line 52 it reads like

if modern geodesy we can deform the crust), strange use of technical terms (e.g. line 13 "small-dip" single plane fault instead of shallow dip), very strange use of adverbs and conjunctions (e.g. line 29 "from" instead of "of"), and even subordinates sentences without verbs. Due to the level of English, the concepts within the text are very hard to understand and I am wondering if some of the largely negative comments I have on the scientific content are indeed related to this problem. From a scientific point of view, although the paper present results really relevant to natural hazards, the reason why the paper was submitted to this journal is never stated (it seems that the only problem is to figure out if the dip end of the seismic rupture is 30 or 50km deep without any explanation about the why we care (despite the large implication in the evaluation of the seismic hazard). The inversion scheme is not completely justified (single flat surface) nor the resolution of the inversion is analyzed. The use of a flat surface also has implication in the analysis of the coulomb stress vs aftershock location (more on this later). The discussion and conclusions make me worry that the authors have not fully understand the analysis they are doing (is it really a big results that using ascending and descending data improving the inversion? It is very well known that the use of ascending and descending data provide a full 3d displacement field while the use of only one of the two provide at most 2d displacement and more likely only line of site deformation). The paper is missing in one of the most important aspect of the use of sentinel wide swath. As explained on the text the use of wide swath does allow observations of the near and far field in a single image but it presents lots of challenges that are not explained in the text at all (I was hoping that I was missing supplemental material!). I realize that this paper was submitted before the paper of Grandin et al (2016, doi:10.1002/2016GL067954) but it is interesting to note that just last week they published in GRL an analysis of the technical challenges to process sentinel data exactly for the same event while this explanation is completely missing in the present manuscript. I am pretty sure that the authors are aware of these chalanges since the results in this manuscript are very similar to the one of Grandin et al. but no mention of them was made in the current version of the paper. To conclude I want also to point out

that the results of the Coulomb stress calculation are biased by the choice of a single flat fault plane in the fault slip inversion (explained more later).

Principal criteria review -Scientific Significance Does the manuscript represent a substantial contribution to the understanding of natural hazards and their consequences (new concepts, ideas, methods, or data)? 3 Fair. The reason why the results are useful for seismic hazards are even not touched a single time in the paper.

-Scientific Quality Are the scientific and/or technical approaches and the applied methods valid? Are the results discussed in an appropriate and balanced way (clarity of concepts and discussion, consideration of related work, including appropriate references)? 3 Fair. Apart from the lack of description of the methodology to process the data from this new satellite, the paper is missing completely an explanation of the resolution of the fault slip inversion, an explanation of why the simplification of a simple single plane geometry for the fault is sufficient (I think it could but then one would get the problem showed in the Coulomb stress calculation). Furthermore the last part of the paper fail in recognizing that the approximation of a bending subduction plane with a flat surface bias the location of the aftershock with respect to the selected fault plane.

Presentation Quality Are the scientific data, results and conclusions presented in a clear, concise, and well-structured way (number and quality of figures/tables, appropriate use of technical and English language, simplicity of the language)? 4 Poor. I have already explained the problem with the English but also thing like presenting the interferogram as a phase figure instead of the unwrapped displacement make the paper very hard to understand.

In conclusion I do not think if the paper should be rejected or be reconsidered after major revisions.

More detailed review points:

Line 11 and line 124 (and I think in other points). What is the meaning of half circle

convex to the east? First you do not have the full displacement since the deformation in the west area is masked by the sea. Second a point source would always give a "circular" area of deformation. Do you want to say that the deformation is not elongated in the along strike direction (that is an interesting observation suggesting a small aspect ratio between length and width of the fault)

Line 13 You can have small angle dip or shallow fault but not small-dip fault

Line 29 What is the meaning of a huge earthquake? you should avoid to use term like huge big small since are all relative terms. For example the 2015 "huge" earthquake is pretty small with respect to the 1960 event. From is the wrong word Line 30 Take away of which More than say "at the latitude. . .." I would say at the location of the earthquake.

Line 32 Why "begins" the subduction? The subduction started at least 40Myr ago and definitely does not start geographically in this location. . .

Lines 34-40 Please rewrite the full sentence. Try to use less subordinates, and be more descriptive. Also put the references in the correct position in the sentence. If the meaning of the sentence allows it put the references at the end.

Line 40 In a statement like this you should specify from when to wen

Figure 1 More than the epicenters of the past events it would have been nicer the area of rupture (it could be derived by many publications, eg the referenced one of Vigny). Some text is not readable (e.g. "South Amarican plate" or Chile trench). Dots for aftershock andsimbles for cities are too similar.

Line 51 Why it is important to understand the subduction zone? Here it would be a perfect place to explain why it is important for natural hazards

Line 52 I think "obtain" is the wrong verb. It sounds like if geodesy is deforming the crust.

Line 55 Which one is "this issue"

Line 61 remove great

Line 64 remove both the, and downloaded (I suppose that if you process the data you obtain them somehow)

Line 66 You do not have "three different constraints" but you do three different inversions of three different dataset

Line 68 Why additionally?

Line 70 this is the main point that make the paper possibly important!

Line 86 I would say postseismic deformation more than aftershock deformation. There are multiple processes that can lead to postseismic deformation and afterslip is only one of them (and also not entirely explained by seismic deformation).

Line 97 What do you mean by many times? What are you really doing to do this critical step? How many times? Are you using a montecarlo method (if I read many times I would assume that). I am wondering if the jump visible in the residuals (panel I, J, and L in figure 2) are related to problems in this process.

Figure 2 Why not unwrapping the images? From the phase image for example I can not see in any way what you state in line 132.

Line 123 While is not the correct word, probably when will be more appropriate

Line 124 Half circle convex is a pretty bad description! And does not means anything

Line 126 why within??

Line 126-132 needs to be completely rewritten it is very hard to understand. In particular since the unwrapped deformation is not presented in any figure.

Line 138 How do you see from focal mechanism that the surface trace closely follows the trench axis???

Line 138 Is a single fault plane a good approximation. It could be but it would strongly

bias the determination of the lowest point of slip on the fault plane. In particular if like in this region the Benioff-Wadati plane (thus likely the slab itself)seems very much bending and the slab in this part of the trench is not a shallow dipping slab.

Line 136-164 (VERY IMPORTANT!!!) Since your results are influenced very strongly by the choice of the parameterization of your model (thus the taxellation of your plane or the size of the patches), and by the selected smoothing (beta) you MUST explain how do you select the best smoothing factor and how good is the resolution of your model. Without this explanation the results are essentially meaningless, in particular regarding the depth of slip on the fault. I need to say that the paper of Melgar et al 2016 (also out the past week on GRL) obtain from seismic and geodetic data a similar slip pattern than the one found in this manuscript, suggesting the results be correct. Another very important point is if the resolution (and best smoothing) is the same for all 3 inversions.

Line 153 "half space model using Okada" add (1985) at the end of line

Line 159 Is the rake fixed for all patches or every patch can have a different rake and the range is the value for the different patches.

Line 160 "to the surface" I think "to the trench" would be more correct.

"steadily modified" what is the meaning of this? Which method did you use to modify the parameters?

Line 162 How does this value compare with slab dip from models like slab1 (Hayes et al 2012)?

Line 164 What kind of resolution test did you make? Any results to show? It seems to me that 10km resolution at depth 50km could be to high resolution. . . (but it is possible to obtain it, if this is the case it needs to be shown).

Line 182 It seems you should have enough point to constrain the deformation also for the ascending data alone pretty well. I am wondering if the problem is the unwrapping

and the fact that the far field within your image is not really at 0 displacement (thus you get smaller displacement at surface than the real one. I am also wondering if the optimal smoothing in this inversion is different from the optimal smoothing in the other inversions

Line 185 It seems to me that the area of slip from the ascending data only is much smaller and the slip is really smaller so it seems strange that the 2 magnitudes are so similar (unless the color scale for the figure 3 is pretty bad and the slip and area of slip are after all not so different).

Line 192 Why did you use the same weight for the ascending and descending in the combined inversion? What does happen if the two weight are different?

Line 193 Symmetric with respect to what?

Line 197 Not a big surprise! The combined dataset allow you to study the full 3d deformation at the surface (or if we assume that northsouth deformation is not so well constraint in the wide swath, at least a full 2d deformation! Not a surprise it is defined better the fault slip. It would have been nice to see a map of the unwrapped deformation from ascending, descending and combined.

Figure 3 It is very hard to read. I would take away topo-bathy and have a better colorscale (for example going to a light color where you do not have deformation).

By the way the paper of Melgar et al in GRL show the presence of different patches with higher slip. I am wondering if your results would also have them with different smoothing and/or different colorscale or your resolution is not good enough to have such patches.

Line 210 It seem that you are not using your fault slip but the one from Lin 2004 and Toda 2004. Be sure to put the reference in the correct place in the sentence

Line 212 it is not that have great influence in earthquake activity but that can trigger seismicity already ready to go. I would rewrite this sentence.

Line 214 (very important) Assuming a single flat fault plane, your model does only approximate the geometry of the plate boundary interface or of the slab. As for the focal mechanism the best plane you will get is mainly influenced by the area with the largest slip (thus shallow). Since the slab in this area is not shallow dipping it is clear that the slab surface would tend to be lower than the one of the fault plane you are inverting for. THIS DOES NOT MEAN THAT SLIP ON YOUR PLANE INCREASES THE LIKELYHOOD OF EARTHQUAKE DEEPER THAN YOUR PREFERED SLIP MODEL! Figure 4b is perfectly compatible with the Benioff Wadati plane in the area. This is why before I was asking a comparison with Slab1! Probably your slip inversion should have been done on a surface following the seismicity more than on a flat surface. This is the real meaning of your figure 4b! By the way it is also important to point out that the location of the aftershock in the figure is from teleseismic and not relocated!

Line 219 In figure 4a it looks like if you have more events in the blue areas than in the red areas. You state that your computed Coulomb stress correlate very well with seismicity distribution. How do you compute the correlation? I am wondering if the seismicity in the blue area is in reality is around patches that did not rupture during the main shock as indicated by Melgar et al. (2016).

Figure 4 A is the seismicity window for depth? B I can not see the blue line but I think the fault interface more than be a line is a curved plane. C it would be great to have seismicity also in this figure.

Line 247 You must show resolution tests!

Line 270 "half circle"????

Line 270-277 I can not understand what you are discussing here. Half circles, NS symmetric, connective rupture? No clue… By the way I am not expecting the subduction of nz uder sa to behave the same along the trench since there are huge differences in things like slab dip! How do you know about barrier or locking, coupling? You have only coseismic data not pre-seismic! What does your paper says about segmentation? I am

pretty sure that a Mw8 would not have uniform slip without any barrier at all (indeed seismic data show significant complexity in the rupture)

Line 284 Not really until you show the resolution tests

Line 297 more than speaking of % of fit it would be nice to give the metric used for the inversion (eg L2)

Line 296-300 I do not agree with this conclusion based on the comments given before.

———————————————

---

## Referee Comment (RC2) · Anonymous Referee #2 · 23 Mar 2016

This paper studies InSAR data collected for the 2015 Chile earthquake, in order to determine the fault geometry, the slip distribution and the related distribution of Coulomb stress change, to be compared with the aftershock distribution. The English usage is very poor along all the paper length. I found difficult to follow the description of the work made on INSAR observation, but I am not a specialists of the subject. The motivations of the study, the choice of the inverted data together with their advantages, and the impact of results are not enough discussed. Some specifications are not given for result reproducibility. Some results are not interpreted correctly.

Scientific Significance: 3 Fair. Currently, elastic half space inversions and computation of Coulomb stress changes for planar dislocations requires almost standard techniques. Unfortunately, another paper recently provided a more resolved information about this earthquake, by jointly inverting also other kinds of data and using a more complex geometry of fault (curved surface) and elastic structure (Melgar et al. 2015 doi:10.1002/2015GL067369).

Scientific Quality: 3 Fair. Results of the Coulomb stress analysis (see below) can be obtained and interpreted more correctly. However, concerning the slip distribution (the extent along dip, the shallow elongation along strike and the relative location with respect to the main-shock hypocentre) the present results are similar to the ones obtained by Melgar et al. (2015). Likely due to the "equal weight" (lines 191-192), results of the joint inversion (ascending plus descending INSAR data) do not differ significantly from those obtained using only descending data. At the same time, model residuals (Figure 2 I-L) are not discussed, so that the joint inversion is not completely justified.

Presentation Quality: 4 Poor. Besides the poor English usage, the LOS displacement increments shown in figure 2 are scarcely interpretable. Some of the used parameters are not specified. Aftershock hypocenter locations should be evidenced also in cross sections together with the rupture extent (Figure 4c). The same saturating values of Coulomb stress (min/max in the colour palette) should be used both in Figure 4a and 4c. The two tables can be more comprehensively organized.

To be publishable, the paper should improve the presentation and compare its results with that obtained by Melgar et al (2015), with trying to interpret the differences in the light of the different resolving power of the data used and the different modelling assumption made.

Detailed comments:

Line 70, 212 and 218 "shear" -> "Coulomb"

Line 142: "Firstly": Before making the linear inversion for the slip distribution, authors make the nonlinear joint inversion (of both ascending and descending data) for the fault

geometry (optimal model ,results shown in Table 2). Unlike the inversion for the slip distribution, in the nonlinear inversion authors do not consider separately ascending and descending InSAR data.

Line 151-152 I agree with referee 1: the criterion used to choose the smoothing factor ïĂÍ\beta) and its chosen value should be declared.

Line 160 "is to the surface" -> "is put at the surface of the elastic half-space".

Table 2 misses the average value of slip and rake assumed or estimated. Parameters fixed or estimated should be distinguishable in Table 2. In Table 2, rather than in Table 3, I would suggest the authors to compare the results of the optimal model with evidences from USGS and GCMT.

Table 3: it is necessary to declare the shear modulus (or rigidity) value used to estimate the seismic moment, as reported in the last three rows. On the contrary, here reporting the same data concerning dip and strike as in Table 2 is unnecessary. Reporting the maximum slip together with the depth of the down-dip edge of the rupture, according to the different data sets, should help readers in understanding how the inferred results depend on the particular data set. Please check the rake angle estimated with descending data which is declared as $110°$ at line 177.

Lines 178, 185 and 194. I am surprised that the "fitting degree" (not defined) is so high, giving the results shown in Figure 2 I-L.

Line 181. What is the "scope" of the slip magnitude? Concerning the lower slip values or seismic moment estimated with ascending data, referee 1 gives an interpretation more articulated and convincing, than the one given by the authors. The displacement observed at a GPS station is a useful constraint to solve for the true displacement observed in the LOS direction.

Line 211. Stain->Stein. I agree with referee 1: likely aftershocks appear below the fault, because the true fault curvature is neglected. Often, the distribution of the seismicity

hypocentres, possibly relocated, in a vertical cross section, allows us to delineate the true dip of the mainshock fault.

Line 214. In general, the distribution of aftershock is not used to choose the "receiver fault mechanism". If the concern are aftershocks, the best thing to do is considering their focal mechanism in order to determine the" mechanism of the receiver fault". As said, aftershock alignments suggest the geometry of the "source fault" (where the mainshock occurred), therefore choosing this geometry for the "receiver fault" coincides with assuming that aftershocks occurred on faults with the same fault mechanism as the source fault. We cannot state that the authors chose this last strategy because in this paper the source fault does not have the same dip as inferred from aftershock alignments.

Line 218-222: The following two statements are scarcely supported by Figure 4 results: 1) "(we) find aftershocks (depth in 20km-30km) locations correlate well with the areas having increased Coulomb stress", 2) "most areas with increased Coulomb stress appeared beneath the main shock fault plane, which is consistent with the location where aftershocks took place."

1) Actually in Figure 4a the majority of aftershocks seems to be shadowed (negative coulomb stress change) by the main rupture. This suggest that the 30 km of depth of the map view is above the down-dip edge of the rupture at least close to line B-B' (as also stated at lines 19, 278, 295). If this is true, the positive Coulomb stress values within the horizontal projection of the fault rupture likely are not due to the slip distribution, given the absence of asperities (regions of no slip) within the rupture surface, as evident from Figure 3c. In obtaining this result, a role may have the change in the receiver dip with respect to the source dip (see last point), or numerical problems due to fault discretization near the fault plane, evident mainly in cross sections (Figure 4c).

2) In Figure 4c, below the fault plane, the most reliable positive feature is the off-fault lobe of Coulomb stress, which is located at a distance of about 200 km. It is due

to tensile stress changes caused by the main rupture (the so-called antithetic lobe). However few of the aftershocks reported in Figure 4b seem to locate there.

Lines 273-275 Sentence to be rewritten for clarity.

---

## Author Comment (AC1) · 26 May 2016

Dear editors and reviewer(s), thank for your comments and suggestions. Replies as follows:

The paper "Coseismic deformation field derived from Sentinel-1A data and slip inversion of the 2015 Chile Mw8.3 earthquake" present surface deformation associated with the past year Chilean earthquake evaluated using the new ESA satellite Sentinel 1-Ain wide swath mode. The data are then modeled with a very simple (probably oversimplified) model using an elastic half space and simulating the fault plane as a single flat surface. The fault slip computed by this inversion is then used to compute Coulomb failure stress and compared it to the aftershock distribution.

The paper, in particular the last two part of it is very problematic from a scientific point of view. The English of the full paper need major reworking, with presence of many colloquialisms (eg. Line 29"huge" earthquake), sentences that do not make any sense (e.g. line 52 it reads like if modern geodesy we can deform the crust), strange use of technical terms (e.g. line13 "small-dip" single plane fault instead of shallow dip), very strange use of adverbs and conjunctions (e.g. line 29 "from" instead of "of"), and even subordinates sentences without verbs. Due to the level of English, the concepts within the text are very hard to understand and I am wondering if some of the largely negative comments I have on the scientific content are indeed related to this problem.

**Answer:**

We will check up the text of the manuscript carefully, and correct grammatical errors and usage errors, finally we will asked for a native speaker of English to read the revised manuscript and adjust some expressions.

From a scientific point of view, although the paper present results really relevant to natural hazards, the reason why the paper was submitted to this journal is never stated (it seems that the only problem is to figure out if the dip end of the seismic rupture is 30 or 50km deep without any explanation about the why we care (despite the large implication in the evaluation of the seismic hazard).

**Answer:**

(1) This journal is an authoritative magazine on natural disasters, I have read some articles about the earthquake disaster published in the journal article, so I want to contribute my article to this magazine.

(2) The significance of this article is to find out the characteristics of the surface deformation field and fault rupture of this Chile earthquake, and provide the basis for seismic disaster assessment and analysis of earthquake risk in the future in this region. In the revised version, we will add more relevant content.

The inversion scheme is not completely justified (single flat surface) nor the resolution of the inversion is analyzed. The use of a flat surface also has implication in the analysis of the coulomb stress vs aftershock location (more on this later). The discussion and conclusions make me worry that the authors have not fully understand the analysis they are doing (is it really a big results that using a scending and descending data improving the inversion? It is very well
known that the use of ascending and descending data provide a full 3d displacement field while
the use of only one of the two provide at most 2d displacement and more likely only line of site
deformation).
**Answer:**
(1) The work of this paper is done at the early stage of the earthquake. We use only S1A
ascending and descending data to invert the fault slip distribution. In the inversion, the
sensitivity of slip distribution to the fault geometry is related to the used data type.
When there is only InSAR data used, the influence of the curved and planar faults on
the slip distribution is not very large. Zhang (2015) made a meaningful discussion about
this      question      in      his      Article      in      Seismological      Research      Letters
(http://srl.geoscienceworld.org/content/86/6/1578). In view of this, we used the plane
fault.
(2) We will rewrite the discussion and conclusions based on the revised manuscript and
reviewer's comments, and more deeply and clearly express the conclusion and
significance of our paper.
(3) In the modified version, we will add a resolution test and analysis.
The step as follows: after we set up the fault model, we carried out the resolution test
(Figure S1.). Firstly, we construct a new fault slip model, and use the initial fault slip to
determine InSAR observation in LOS direction. Then we use this InSAR data from
forward modeling to invert fault slip. By comparing the initially constructed fault slip
with the inverted one, we find that the resolution in the shallow portion (0-150km along
dip) is good. While in deeper portion the resolution is finite. The magnitude of slip in
deeper zones is between 0m and 1m. Our fault slip model constraint by InSAR data sets
is limited to the determination of deep slip. 150km along dip is equivalent to about
50km in depth (dip=18.3°), that is to say, the fault slip between 0km to 50km in our
model is reliable.

[Figure]

**Figure S1.** (a) The initial fault slip. (b) The inverted fault slip by our model. Red color
indicate the slip value is 0m, and pink stands for slip value 1m.
(4) Indeed, the joint use of the ascending and descending InSAR data has better
constraints on the interpretation of the deformation field and inversion of fault slip.
Generally,the more geodetic data set used, the better fault slip can be obtained.
In addition, by using the the ascending and descending data, we calculate the vertical
displacement and east–west (E-W) displacement (Figure S2), this will help us to better
understand the earthquake induced ground displacement. Figure S2a is E-W component.

The maximum displacement to west is about 2.2m. Figure S2b is vertical component. The maximum uplift is about 0.2m and the maximum subsidence is about 0.2m too.

[Figure]

**Figure S2.** Displacement components computed from the descending and ascending deformation maps (a) displacement component in east−west; (b) displacement component in vertical.

The paper is missing in one of the most important aspect of the use of sentinel wide swath. As explained on the text the use of wide swath does allow observations of the near and far field in a single image but it presents lots of challenges that are not explained in the text at all (I was hoping that I was missing supplemental material!). I realize that this paper was submitted before the paper of Grandin etal (2016, doi:10.1002/2016GL067954) but it is interesting to note that just last week they published in GRL an analysis of the technical challenges to process sentinel data exactly for the same event while this explanation is completely missing in the present manuscript. I am pretty sure that the authors are aware of these challanges since the results in this manuscript are very similar to the one of Grandin et al. but no mention of them was made in the current version of the paper.

**Answer:**
The work of this paper is done at the early stage of the earthquake. In the new data processing, we really address some of the problems and get good results. In the modified version, we will add the introduction to new data processing methods and technology.

To conclude I want also to point out that the results of the Coulomb stress calculation are biased by the choice of a single flat fault plane in the fault slip inversion (explained more later).

**Answer:**
I agree with the comments of the reviewers. The enhanced area of Coulomb stress change is below the main rupture fault, and aftershocks, especially those below 30km are also located under the rupture fault, this indicates the fault dip probably changes steep in deep depth, maybe our used flat fault leads to the deviation. In the modified version, we will try bending fault and re-calculate the Coulomb stress changes, and give a reasonable explanation.

Principal criteria review -Scientific Significance Does the manuscript represent a substantial contribution to the understanding of natural hazards and their consequences (new concepts, ideas, methods, or data)? 3 Fair. The reason why the results are useful for seismic hazards are even not touched a single time in the paper.

**Answer:**

The purpose of this article is to find out the characteristics of the surface deformation field and fault rupture of this Chile earthquake, and provide the basis for seismic disaster assessment and analysis of earthquake risk in the future in this region. In the revised version, we will add more relevant content.

Scientific Quality Are the scientific and/or technical approaches and the applied methodsvalid? Are the results discussed in an appropriate and balanced way (clarity ofconcepts and discussion, consideration of related work, including appropriate references)?3 Fair. Apart from the lack of description of the methodology to process the data from this new satellite, the paper is missing completely an explanation of the resolution of the fault slip inversion, an explanation of why the simplification of a simple single plane geometry for the fault is sufficient (I think it could but then one would get the problem showed in the Coulomb stress calculation). Furthermore the last part of the paper fail in recognizing that the approximation of a bending subduction plane with a flat surface bias the location of the aftershock with respect to the selected fault plane.

**Answer:**

(1) During the new data (S1A TOPS mode) processing, we really address some problems and get good results. In the modified version, we will add the introduction to the new data and its processing methods and technology.

(2) In the modified version, we will add a resolution test and analysis. As described in the previous page.

(3) About the use of a flat plane fault in our inversion, see explanation described in the previous page.

(4) Thank you for the reminder that flat plane fault causes the bias of the location of the aftershock. We will give corresponding explanation in the discussion in new version.

Presentation Quality Are the scientific data, results and conclusions presented in a clear, concise, and well-structured way (number and quality of figures/tables, appropriateuse of technical and English language, simplicity of the language)? 4 Poor. I have already explained the problem with the English but also thing like presenting the interferogram as a phase figure instead of the unwrapped displacement make the paper very hard to understand. In conclusion I do not think if the paper should be rejected or be reconsidered after major revisions.

**Answer:**

We will make a major revisions according to the comments of the reviewers. We will redraw some of figs, such as interferogram, which will be expressed as unwrapped displacement instead of a phase figure, and we will re-organize and arrange the contents of the article, and we will ask for some person who are familiar with the English to modify the final text in modified version.

**More detailed review points:**

Line 11 and line 124 (and I think in other points). What is the meaning of half circle convex to the east? First you do not have the full displacement since the deformation in the west area is masked by the sea. Second a point source would always give a "circular" area of deformation. Do you want to say that the deformation is not elongated in the along strike direction (that is an interesting observation suggesting a small aspect ratio between length and width of the fault)

**Answer:**
Here, we would like to describe the shape of the deformation field observed on land, "half circle convex to the east" means the observed coseismic deformation field on land looks like semicircle, convex to the east. As reviewer said that suggesting a small aspect ratio between length and width of the fault. We will improve our English expression in in modified version.

Line 13 You can have small angle dip or shallow fault but not small-dip fault

**Answer:**
We will corrected it in the revised manuscript.

Line 29 What is the meaning of a huge earthquake? you should avoid to use term like huge big small since are all relative terms. For example the 2015 "huge" earthquake is pretty small with respect to the 1960 event. From is the wrong word Line 30 Take away of which More than say "at the latitude: : :." I would say at the location of the earthquake.

**Answer:**
We will modify these inappropriate expressions in the revised manuscript.

Line 32 Why "begins" the subduction? The subduction started at least 40Myr ago and definitely does not start geographically in this location.

**Answer:**
It has been modified in the revised manuscript

Lines 34-40 Please rewrite the full sentence. Try to use less subordinates, and be more descriptive. Also put the references in the correct position in the sentence. If the meaning of the sentence allows it put the references at the end.

**Answer:**
We will complete these changes in the modified version.

Line 40 In a statement like this you should specify from when to wen

**Answer:**

It has been modified in the revised manuscript.

Figure 1 More than the epicenters of the past events it would have been nicer the area of
rupture (it could be derived by many publications, eg the referenced one of Vigny). Some text
is not readable (e.g. "South Amarican plate" or Chile trench). Dots for aftershock and simbles
for cities are too similar.

**Answer:**
We will modify the Fig1 according to requirements of reviewers in the revised
manuscript.

Line 51 Why it is important to understand the subduction zone? Here it would be a perfect
place to explain why it is important for natural hazards

**Answer:**
Studying the down dip limit of seismogenic rupture may provide insights into the
rheological controls on the earthquake process, and also provide clues to understand the
relationship between the down dip limit of stick slip behavior and the depth of the
continental Moho at its intersection with the subduction interface.
Existing research shows that plate boundaries can be divided into three main zones as
the depth increase: an aseismic up dip zone, the seismogenic zone, and a deep aseismic
zone. Identifying the transitions between these zones and the processes controlling their
locations are key goals in understanding the mechanics of slip along subduction zones.
Although we have a general understanding of the processes that occur in the subduction
zones, many details remain obscure. For example, we know that earthquakes reflect the
rapid release of strain associated with prior locking of the shallow plate interface and
strain accumulation during interseismic periods lasting tens to hundreds of years. The
reason why some interpolate subduction earthquake are relatively modest in size,
rupturing relative small areas with limited along strike (trench-parallel) rupture length
(<100km), while others, such as the great 1960 Chile earthquake rupture more than
1200km along strike, are still uncertain.
Here we use nearly complete coverage from S1A data to resolve the spatial variations
of the seismogenic fault slip, and thus provide tight constraints on the depth of this
rupture and transitions between seismic and aseismic zones

Line 52 I think "obtain" is the wrong verb. It sounds like if geodesy is deforming the
crust.

**Answer:**
It is really not a proper word, we will use "measure" instead of "obtain" in the revised
manuscript.

Line 55 Which one is "this issue"

**Answer:**
"This issue" is refer to the down-dip limit of the seismogenic zone.

Line 61 remove great

**Answer:**

It has been modified in the revised manuscript.

Line 64 remove both the, and downloaded (I suppose that if you process the data you obtain them somehow)

**Answer:**

It has been modified in the revised manuscript.

Line 66 You do not have "three different constraints" but you do three different inversions of three different dataset

**Answer:**

It has been modified in the revised manuscript.

Line 68 Why additionally?

**Answer:**

We have deleted "additionally" in the revised manuscript.

Line 86 I would say postseismic deformation more than aftershock deformation. There are multiple processes that can lead to postseismic deformation and afterslip is only one of them (and also not entirely explained by seismic deformation).

**Answer:**

We will change the text description according to reviewer's opinion and references in the revised manuscript.

Line 97 What do you mean by many times? What are you really doing to do this critical step? How many times? Are you using a montecarlo method (if I read many times I

would assume that). I am wondering if the jump visible in the residuals (panel I, J, and

L in figure 2) are related to problems in this process.

**Answer:**

Considering that we need to achieve a high registration accuracy of a very small fraction of an SLC pixel, especially in azimuth direction. We estimate the offset by iteration, until the azimuth offset correction becomes at least smaller than 0.02 SLC pixel. In our work, we performed iteration 3 times. The jump visible in the residuals (panel I, J, and

L in Figure 2) are not related to problems in this process. I think they may be the intersection of different frames (3 frames in descending track and 2 frames in ascending track). We processed each frame separately, and then stitching them together into a complete interferogram.

Figure 2 Why not unwrapping the images? From the phase image for example I can not see in any way what you state in line 132.

**Answer:**

We will redraw Figure 2 expressed as unwrapped displacement easy to see the size of the LOS displacement value in the revised manuscript. The old is rewrapped interferogram with color cycle 12cm.

Line 123 While is not the correct word, probably when will be more appropriate

**Answer:**

It has been modified in the revised manuscript.

Line 124 Half circle convex is a pretty bad description! And does not means anything

**Answer:**

It has been modified in the revised manuscript.

Line 126 why within??

**Answer:**

We have changed "within" to "about"

Line 126-132 needs to be completely rewritten it is very hard to understand. In particular since the unwrapped deformation is not presented in any figure.

**Answer:**

We will completely rewrite these sentences according to the remapped Figure 2

(expressed as unwrapped displacement).

Line 138 How do you see from focal mechanism that the surface trace closely follows the trench axis???

**Answer:**

Sorry, my expression is not clear here. We will change this expression in the revised manuscript. In fact, we determined the fault geometry according to focal mechanism and Inferred the top edge of the seismogenic fault follows the trench closely.

Line 138 Is a single fault plane a good approximation. It could be but it would strongly bias the determination of the lowest point of slip on the fault plane. In particular if like in this region the Benioff-Wadati plane (thus likely the slab itself)seems very much bending and the slab in this part of the trench is not a shallow dipping slab.

**Answer:**

We agree with the comments of the reviewers. We have realized that a flat fault approximation in our inversion does lead to a deviation from the aftershock and the deviation of the lowest point of slip on the fault plane from the actual position. We will take new inversion using bending fault plane in the revised manuscript.

Line 136-164 (VERY IMPORTANT!!!) Since your results are influenced very strongly by the choice of the parameterization of your model (thus the taxellation of your plane or the size of the patches), and by the selected smoothing (beta) you MUST explain how do you select the best smoothing factor and how good is the resolution of your model. Without this explanation the results are essentially meaningless, in particular regarding the depth of slip on the fault. I

need to say that the paper of Melgar et al 2016 (also out the past week on GRL) obtain from seismic and geodetic data a similar slip pattern than the one found in this manuscript, suggesting the results be correct. Another very important point is if the resolution (and best smoothing) is the same for all 3 inversions.

**Answer:**

(1) The choice of the best smoothing factor in our inversion is through a model of the misfit and roughness trade-off curve (Figure S3). When the smooth factor value increase, the misfit value would increase, while the roughness would decrease. By using a trade-off curve, we find that smoothing factor 0.3 can best fit InSAR descending data.

Because constraint degree by smooth factor in fault slip inversion for ascending and descending data is similar, we select same smooth factor 0.3 in three inversion.

[Figure]

**Figure S3**    The trade-off curve between roughness and misfit

(2) We have read the paper of Melgar et al published on GRL and other published papers, during the modification of our manuscript, we will refer to these articles.

(3) To assess the resolution capabilities of the fault grid, we conducted a test in version with six different grid sizes of the fault constrained by Sentinel-1A ascending and descending data jointly. We divide fault plane as rectangular patches with length 5×5

km, 10×10 km, 20×20 km, 30×30 km, 40×40 km, 50×50 km (Figure S4). The smaller the fault patches, the more clearly the sliding feature, but takes more time. We find that

When the mesh size is 5km and 10km, the resultant slips are similar, but the computation time is longer for smaller grid size. When the patch size changes from 20

×20km to 50×50km,the maximum fault slip becomes smaller, respectively , in turn

7.7m, 7.52m, 7.06m, 6.41m. The depth of slip area are all over 60km, which is different from other results published at present. So we chose grid patch size as 10km×10km in our inversion.

[Figure]

[Figure]

**Figure S4.** Fault slip resolution test with different fault patch size in the inversion constrained by Sentinel-1A ascending and descending data jointly. The fault patch size is (a) 5 km×5 km, (b) 10 km×10 km, (c) 20 km×20 km, (d) 30 km×30 km, (e) 40 km×40 km, (f) 50×50 km.

Line 153 "half space model using Okada" add (1985) at the end of line

**Answer:**

It has been modified in the revised manuscript.

Line 159 Is the rake fixed for all patches or every patch can have a different rake and the range
is the value for the different patches.
**Answer:**
The rake is fixed for all patches.

Line 160 "to the surface" I think "to the trench" would be more correct. "steadily modified"
what is the meaning of this? Which method did you use to modify the parameters?
**Answer:**
"to the surface" has been modified to "to the trench" in the revised manuscript. When
we modify the parameter, we increase the value gradually. For example, when we set
dip angle, we take trial values between 10° and 30° one by one manually. And find
model of dip=18.3°can best fit observation. Then we change other parameters.

Line 162 How does this value compare with slab dip from models like slab1 (Hayes et al 2012)?
**Answer:**
The flat fault model derived from our inversion reflects a dip angle of 18.3°. And in
slab1.0, the average dip angle of the interface in Central Chile is 16°, the transition
depth between seismic and aseismic zones is about 50km. But considering that the
model is just average with limited accuracy, while the dip angle of the fault may be
steep in the deep part according to aftershocks distribution, so we will take new
inversion using bending fault plane in the revised manuscript.

Line 164 What kind of resolution test did you make? Any results to show? It seems tome that
10km resolution at depth 50km could be to high resolution : (but it is possible to obtain it, if
this is the case it needs to be shown).
**Answer:**
We did the resolution test, see the explanation above.

Line 182 It seems you should have enough point to constrain the deformation also for the
ascending data alone pretty well. I am wondering if the problem is the unwrapping and the fact
that the far field within your image is not really at 0 displacement (thus you get smaller
displacement at surface than the real one. I am also wondering if the optimal smoothing in this
inversion is different from the optimal smoothing in the other inversions
**Answer:**
(1) When unwrapping, we really set the deformation of the far field within the picture
to 0. But just as you said, the actual far-field deformation is not 0 in ascending data due
to incomplete coverage. We will make a careful check to the unwrapped result or re-
unwrap it using different reference point at far field to get a good result in the revised
manuscript.
(2) The optimal smoothing factor in ascending data inversion is the same as that in other
inversions.

Line 185 It seems to me that the area of slip from the ascending data only is much smaller and the slip is really smaller so it seems strange that the 2 magnitudes are so similar (unless the color scale for the figure 3 is pretty bad and the slip and area of slip are after all not so different).

**Answer:**

We agree with you. We are aware of the two question. On the one hand, the much smaller slip value in ascending data inversion is likely to be related to the unwrapping, on the other hand, the color scale we used is not appropriate, so that distinction is not clear. We will seriously examine and revise these issues in order to improve the clarity and quality of the map in the revised manuscript.

Line 192 Why did you use the same weight for the ascending and descending in the combined inversion? What does happen if the two weight are different?

**Answer:**

We think the constraint ability of the ascending and descending data to the inversion is the same, so we used the same weight for the ascending and descending data in the combined inversion. But we will carry out the test with different weights in order to know what happen if the two weight are different in the revised manuscript.

Line 193 Symmetric with respect to what?

**Answer:**

Here we want to say the slip area looks like a semicircle shape slightly elongated in trench direction, and symmetrically distribute to the north and south sides along the trench.

Line 197 Not a big surprise! The combined dataset allow you to study the full 3d deformation at the surface (or if we assume that north south deformation is not so well constraint in the wide swath, at least a full 2d deformation! Not a surprise it is defined better the fault slip. It would have been nice to see a map of the unwrapped deformation from ascending, descending and combined.

**Answer:**

We are in favor of your opinion. We will give a maps of the unwrapped deformation from ascending and descending in the revised manuscript, we will also calculate the vertical and east–west displacement components by combining ascending and descending data.

Figure 3 It is very hard to read. I would take away topo-bathy and have a better color scale (for example going to a light color where you do not have deformation). By the way the paper of Melgar et al in GRL show the presence of different patches with higher slip. I am wondering if your results would also have them with different smoothing and/or different colorscale or your resolution is not good enough to have such patches.

**Answer:**

We will revise the Fig.3 according to the comments of the reviewers. We will use a appropriate color-scale to distinguish the slipping difference to make the fig easy to read. We will compare our new maps with the results of Melgar in the discussion of our new revision. Of course, Melgar et al used many kind of data, including high-rate GPS,
strong motion data, and tide. Our sliding distribution may be different from theirs due
to the use of different data constraints.

Line 210 It seem that you are not using your fault slip but the one from Lin 2004 and Toda
2004. Be sure to put the reference in the correct place in the sentence
**Answer:**
It has been modified in the revised manuscript.

Line 212 it is not that have great influence in earthquake activity but that can trigger seismicity
already ready to go. I would rewrite this sentence.
**Answer:**
It has been modified in the revised manuscript.

Line 214 (very important) Assuming a single flat fault plane, your model does only aproximate
the geometry of the plate boundary interface or of the slab. As for the local mechanism the best
plane you will get is mainly influenced by the area with the largest slip (thus shallow). Since
the slab in this area is not shallow dipping it is clear that the slab surface would tend to be lower
than the one of the fault plane you are inverting for. THIS DOES NOT MEAN THAT SLIP
ON YOUR PLANE INCREASES THE LIKELYHOOD OF EARTHQUAKE DEEPER THAN
YOUR PREFERED SLIP MODEL!
**Answer:**
After reading the comments, we have realized the problems caused by the flat plane
fault approximation in our inversion. The aftershock distribution should reflect the
geometry of the main fault. That is to say the dip angle of the seismic fault is larger in
the deeper portion (about 20 km below). The flat fault model leads to inappropriate
interpretation. In the revised manuscript, we will take new inversion using bending fault
plane and compare with the old one, and give more reasonable interpretation.

Figure 4b is perfectly compatible with the Benioff Wadati plane in the area. This is why before
I was asking a comparison with Slab1! Probably your slip inversion should have been done on
a surface following the seismicity more than on a flat surface. This is the real meaning of your
figure 4b! By the way it is also important to point out that the location of the aftershock in the
figure is from teleseismic and not relocated!
**Answer:**
We totally agree with you. We will use the surface which is consistent with the
aftershock distribution to make a new inversion in our revised manuscript.
The location of the aftershocks in our paper is from teleseismic.

Line 219 In figure 4a it looks like if you have more events in the blue areas than in the red areas.
You state that your computed Coulomb stress correlate very well with seismicity distribution.
How do you compute the correlation? I am wondering if the seismicity in the blue area is in
reality is around patches that did not rupture during the main shock as indicated by Melgar et
al. (2016).

**Answer:**

As mentioned above, under the review of the comments, we have realized that these problems are caused by flat fault model and improper location of the receive fault. We will use bending fault model and another receive fault to improve the result and its interpretation in the revised manuscript.

In addition, similar to Melgar 's result, our inverted slip model also indicates the after shock seismicity mostly occurred at the edge of the main rupture zone(Fig.3).

Figure 4 A is the seismicity window for depth? B I can not see the blue line but I think the fault interface more than be a line is a curved plane. C it would be great to have seismicity also in this figure.

**Answer:**

Figure 4 A stands for Coulomb Failure Stress (CFS) changes at a depth of 30km.

We will redraw these Figs after new inversion based on curved plane fault and again calculates CFS, then we will present new Figs and add seismicity to Fig. C in the revised manuscript.

Line 247 You must show resolution tests!

**Answer:**

As mentioned before, we did it.

Line 270 "half circle"????

**Answer:**

It means semicircle shape.

Line 270-277 I can not understand what you are discussing here. Half circles, NS symmetric, connective rupture? No clue: : : By the way I am not expecting the subduction of nz uder sa to behave the same along the trench since there are huge differences in things like slab dip! How do you know about barrier or locking, coupling? You have only coseismic data not pre-seismic! What does your paper says about segmentation? I am pretty sure that a Mw8 would not have uniform slip without any barrier at all (indeed seismic data show significant complexity in the rupture)

**Answer:**

We are very sorry. This paragraph is really not expressed clearly. We will carefully modify the wording in the new version. Here we would like to compare the coseismic deformation field and the fault slip associated with this earthquake to that of the 2010's Mw8.8 event, discuss the differences between them, and further explain complexity of the subduction zone. Results from Bertrand et al (2010) based on static and high-rate GPS, InSAR and broadband teleseismic data, show that 2010 Mw8.8 earthquake rupture initiated at about 32 km depth and propagated bilaterally resulting in two main slip zones. While for this Mw8.3 event, our result from InSAR ascending and descending joint inversion indicates one slip zone. This may be related to the constraint capacity of different data, may also reflect the complexity and diversity of the subduction zones in different position.

Line 284 Not really until you show the resolution tests

**Answer:**

We did it. We will show it in the revised manuscript.

Line 297 more than speaking of % of fit it would be nice to give the metric used for the inversion (eg L2)

**Answer:**

We will use the metric representation rather than a percentage in the revised manuscript.

Here the fitting degree is defined as follows:

$$Corr = \frac{\sum_{n=1}^{N} O_n \cdot P_n}{\sqrt{\sum_{n=1}^{N} O_n^{2} \cdot \sum_{n=1}^{N} P_n^{2}}}$$

'*Corr*' is the fitting degree, 'n' is the data points index, 'N' is the total sampled points in our inversion. 'O' is the observation data. 'P' is the prediction data. When $O_n \approx P_n$,

$Corr \approx 1$.

Line 296-300 I do not agree with this conclusion based on the comments given before.

**Answer:**

We totally agree with you. We will make a major revision to the manuscript as stated above, and then we will rewrite the conclusion.

---

## Author Comment (AC2) · 26 May 2016

Dear editors and reviewers, thank for your comments and suggestions.

Replies for anonymous referee #2 as follows:

This paper studies InSAR data collected for the 2015 Chile earthquake, in order to determine the fault geometry, the slip distribution and the related distribution of Coulomb stress change, to be compared with the aftershock distribution. The English usage is very poor along all the paper length. I found difficult to follow the description of the work made on INSAR observation, but I am not a specialists of the subject. The motivations of the study, the choice of the inverted data together with their advantages, and the impact of results are not enough discussed. Some specifications are not given for result reproducibility. Some results are not interpreted correctly.

**Answer:**

(1) We will make a major revision of the manuscript, including redraw Fig.2 expressed as unwrapped displacement instead of phase interferograms, redraw Fig.3 to be clear using good color scale, make another inversion based on curved fault, recalculate Coulomb stress change and redraw Fig.4, add resolution test and strengthen the analysis of the content of the discussion, and so on.

(2) We will asked for a native English speakers to modify the finally revised the manuscript text.

Scientific Significance: 3 Fair. Currently, elastic half space inversions and computation of Coulomb stress changes for planar dislocations requires almost standard techniques. Unfortunately, another paper recently provided a more resolved information about this earthquake, by jointly inverting also other kinds of data and using a more complex geometry of fault (curved surface) and elastic structure (Melgar et al. 2015doi:10.1002/2015GL067369).

**Answer:**

(1) Melgar et al. really made a very excellent research in this Chile's earthquake (Melgar et al. 2015doi:10.1002/2015GL067369), we have read his paper carefully. He used four kinds of data (high-rate GPS records, strong motion records, two interferograms from ascending and descending of the S1A satellite, and tide gauge records), jointly inverted the fault slip model. They found that this earthquake produced deep and shallow two sliding zones. The "deep" asperity is well separated from the "shallow" asperity by a gap of reduced slip. The deep slip patch extends to 45 km depth, with ~10m peak slip at ~30 km depth. The shallow slip patch ruptured all the way to the trench, with ~10m peak slip at~15 km.

(2) Usually, there are some differences in the slip models constrained by different data sets, although it is necessary to use a variety of data sets to get a good slip model. In fact,dense InSAR data can provide good constrain to the near field deformation. In our paper, we used ascending and descending S1A InSAR data to get the coseismic deformation field and invert the slip model. We obtained one slip zone with maximum slip about 8m at ~10km depth, the slip depth reached 45km, while the large slip occurred in shallow portion. These results are similar to that of Melgar's, In addition to his discovery of the two slip areas. Our results are similar to those obtained by Giuseppe Solaro et al.(Giuseppe Solaro et al. 2016,doi:10.3390/rs8040323 ) using S1A data, which indicates our results are correct.

(3) In addition, in our manuscript, we calculated and analyzed the Coulomb Failure

Stress (CFS) changes and its relation to aftershocks. We also calculate the vertical displacement component and east–west(E-W) displacement component by using the ascending and descending data.

(4) We think these results from our study are significant for understanding the fracture behavior of the earthquake. We will improve and perfect this manuscript according to your comments and suggestions in the modified version.

Scientific Quality: 3 Fair. Results of the Coulomb stress analysis (see below) can be obtained and interpreted more correctly. However, concerning the slip distribution (the extent along dip, the shallow elongation along strike and the relative location with respect to the main-shock hypocentre) the present results are similar to the ones obtained by Melgar et al. (2015). Likely due to the "equal weight" (lines 191-192), results of the joint inversion (ascending plus descending INSAR data) do not differ significantly from those obtained using only descending data. At the same time, model residuals (Figure 2 I-L) are not discussed, so that the joint inversion is not completely justified.

**Answer:**

(1) We are in favor of your opinion. After reading the comments of the two reviewers, we have realized that a flat fault approximation in our inversion lead to a deviation from the aftershock, at the same time, the results of Coulomb Failure Stress (CFS) changes calculated based on the flat fault model are also affected. We will take new inversion using bending fault plane and recalculate the Coulomb Failure Stress (CFS) changes,and give more correct interpretation in the revised manuscript.

(2) We think the reason for the difficulty in distinguishing the results of joint inversion from those obtained using only descending data is due to bad color scale. We will use an appropriate color-scale to distinguish the slip differences of different data sets to make the Fig.3 easy to read in the modified version.

(3) The residual in Figure 2 (I - L) is ~15cm, compared with the coseismic displacement more than 130cm, this residual is acceptable. The residual from inversion of ascending or descending data alone is slightly smaller than that from inversion jointly. This may be due to the slip model in joint inversion to meet both ascending and descending data at the same time.

Presentation Quality: 4 Poor. Besides the poor English usage, the LOS displacement increments shown in figure 2 are scarcely interpretable. Some of the used parameters are not specified. Aftershock hypocenter locations should be evidenced also in cross sections together with the rupture extent (Figure 4c). The same saturating values of Coulomb stress (min/max in the colour palette) should be used both in Figure 4a and4c. The two tables can be more comprehensively organized.

**Answer:**

(1) We will invite a native English speaker to modify the final manuscript and adjust some expressions.

(2) We will redraw Figure 2 expressed as unwrapped displacement instead of phase interferograms in order to make it easy to explain, and we will redraw Figure 4a, 4c and modify them according to reviewer's suggestion.

To be publishable, the paper should improve the presentation and compare its results with that obtained by Melgar et al (2015), with trying to interpret the differences in the light of the different resolving power of the data used and the different modelling assumption made.

**Answer:**

As mentioned above, we will take new inversion using bending fault plane and recalculate the Coulomb Failure Stress (CFS) changes, strengthen the content of the discussion section through the comparison with that of Melgar et al (2015).

**Detailed comments:**

Line 70, 212 and 218 "shear" -> "Coulomb"

**Answer:**

It has been modified in the revised manuscript.

Line 142: "Firstly": Before making the linear inversion for the slip distribution, authors make the nonlinear joint inversion (of both ascending and descending data) for the fault geometry (optimal model, results shown in Table 2). Unlike the inversion for the slip distribution, in the nonlinear inversion authors do not consider separately ascending and descending InSAR data.

**Answer:**

Our fault slip inversion following two steps: firstly, we carry out a nonlinear inversion to constrain the fault geometry, then we perform a linear inversion to retrieve the fault distribution. Different data sources can get different fault geometry in the nonlinear inversion. When we make a nonlinear inversion constrained by ascending data, the obtained fault model has poor fitting to descending data, and vice versa. In order to get a general fault geometry, we use both ascending and descending data in the nonlinear joint inversion.

Line 151-152 I agree with referee 1: the criterion used to choose the smoothing factor (beta) and its chosen value should be declared.

**Answer:**

We select the smoothing factor through trade-off curve model between roughness and misfit which are mutually restricted. When the smooth factor value increase, the misfit value will increase, while the roughness will decrease. By using a trade-off curve (Figure S1.), we find best fitting smoothing factor 0.3 for descending data inversion. Taking into account the degree of constraint by smooth factor in fault slip inversion is similar, the smoothing factor is all set to 0.3 in our three kinds of inversion.

[Figure]

**Figure S1.** The trade-off curve between roughness and misfit

Line 160 "is to the surface" -> "is put at the surface of the elastic half-space".

**Answer:**

It has been modified in revised manuscript.

Table 2 misses the average value of slip and rake assumed or estimated. Parameters fixed or estimated should be distinguishable in Table 2. In Table 2, rather than in Table 3, I would suggest the authors to compare the results of the optimal model with evidences from USGS and

GCMT.

**Answer:**

We will add the corresponding contents to table 2 and table3 according to your opinion in our revised manuscript.

Table 3: it is necessary to declare the shear modulus (or rigidity) value used to estimate the seismic moment, as reported in the last three rows. On the contrary, here reporting the same data concerning dip and strike as in Table 2 is unnecessary. Reporting the maximum slip together with the depth of the down-dip edge of the rupture, according to the different data sets, should help readers in understanding how the inferred results depend on the particular data set.

Please check the rake angle estimated with descending data which is declared as 110₋ at line

177.

**Answer:**

We set the shear modulus $3.0 \times 10^6$ MPa in our fault slip inversion. We will modify table

2 and table 3 to add the corresponding contents mentioned above in the revised manuscript

Lines 178, 185 and 194. I am surprised that the "fitting degree" (not defined) is so high, giving the results shown in Figure 2 I-L.

**Answer:**

(1) The fitting degree is defined as follows:

$$Corr = \frac{\sum\limits_{n=1}^{N} O_n \cdot P_n}{\sqrt{\sum\limits_{n=1}^{N} O_n^{\,2} \cdot \sum\limits_{n=1}^{N} P_n^{\,2}}}$$

'*Corr*' is the fitting degree, 'n' is the data points index, 'N' is the total sampled points
in our inversion. 'O' is the observation data. 'P' is the prediction data. When $O_n \approx P_n$,
$Corr \approx 1$.
(2) The residual error of the observed value minus the model is about 15cm (Figure2 I-
L), compared with the coseismic displacement more than 130cm, this residual is small
and acceptable, so the "fitting degree" is high.
(3) We will redraw the Fig.2 to make the residual value easy to identify.
Line 181. What is the "scope" of the slip magnitude? Concerning the lower slip values or
seismic moment estimated with ascending data, referee 1 gives an interpretation more
articulated and convincing, than the one given by the authors. The displacement observed at a
GPS station is a useful constraint to solve for the true displacement observed in the LOS
direction.
**Answer:**
(1) Here the "scope" of the slip magnitude means the range of the slip area coverage on
fault plane. We will improve the English expression in revised version.
(2) We agree with you. We are aware of the two question after reading the review
comments. On the one hand, the much smaller slip value in ascending data inversion is
likely to be related to the unwrapping, on the other hand, the color scale we used is not
appropriate, so that distinction is not clear. We will seriously examine and revise these
issues in order to improve the clarity and quality of the map in the revised manuscript.
(3) As you said, it is very good to use the displacement observed at a GPS station as
constraint to solve for the true displacement observed in the LOS direction. In the
observation of coseismic deformation, We usually take the far field without deformation
fringes as reference when the GPS is not available in the region of interferogram. That's
what we did in this manuscript. We will check and improve the process of unwrapping
in the new version.
Line 211. Stain->Stein. I agree with referee 1: likely aftershocks appear below the fault, because
the true fault curvature is neglected. Often, the distribution of the seismicity hypocentres,
possibly relocated, in a vertical cross section, allows us to delineate the true dip of the
mainshock fault.
**Answer:**
(1) "Stain" has been modified to "Stein" in the revised manuscript.
(2) We agree with the comments of the reviewers. We have realized that a flat fault
approximation in our inversion does lead to a deviation from the aftershocks. The dip
angle of the seismogenic fault may be greater in the deeper part. So we will take new
inversion using bending fault plane in the revised manuscript, which will improve our
interpretation.

Line 214. In general, the distribution of aftershock is not used to choose the "receiver fault mechanism". If the concern are aftershocks, the best thing to do is considering their focal mechanism in order to determine the" mechanism of the receiver fault". As said, aftershock alignments suggest the geometry of the "source fault" (where the mainshock occurred), therefore choosing this geometry for the "receiver fault" coincides with assuming that aftershocks occurred on faults with the same fault mechanism as the source fault. We cannot state that the authors chose this last strategy because in this paper the source fault does not have the same dip as inferred from aftershock alignments.

**Answer:**
After reading the comments of two reviewers, we have noticed this deviation caused by the flat plane fault approximation in our inversion. The aftershock distribution should reflect the geometry of the main fault. That is to say the dip angle of the seismic fault becomes larger in the deeper portion (about 20 km below). The flat fault model leads to inappropriate interpretation. In the revised manuscript, we will take new inversion using bending fault plane and compare with the old one, and give more reasonable interpretation.

Line 218-222: The following two statements are scarcely supported by Figure 4 results:
1) "(we) find aftershocks (depth in 20km-30km) locations correlate well with the area shaving increased Coulomb stress", 2) "most areas with increased Coulomb stress appeared beneath the main shock fault plane, which is consistent with the location where aftershocks took place."

**Answer:**
We agree with you. The statements in our initial manuscript are really inappropriate due to the deviation from the plane fault approximation. We will make a new inversion using bending fault and recalculate the Coulomb Failure Stress (CFS) changes, redraw the Fig.4, we will add aftershocks to Fig.4C to make the relationship between the CFS changes and the aftershock distribution more clear and easy to judge in our revised manuscript.

1) Actually in Figure 4a the majority of aftershocks seems to be shadowed (negative coulomb stress change) by the main rupture. This suggest that the 30 km of depth of the map view is above the down-dip edge of the rupture at least close to line B-B' (as also stated at lines 19, 278, 295). If this is true, the positive Coulomb stress values within the horizontal projection of the fault rupture likely are not due to the slip distribution, given the absence of asperities (regions of no slip) within the rupture surface, as evident from Figure 3c. In obtaining this result, a role may have the change in the receiver dip with respect to the source dip (see last point), or numerical problems due to fault discretization near the fault plane, evident mainly in cross sections (Figure 4c).

**Answer:**
Under the guidance of the referee's opinions, We do realize that the results shown in Fig.4 and corresponding explanation are not appropriate. Maybe the flat fault model with low dip angle(18.3°) used in our initial manuscript leads to the deviation of the source fault from the distribution of the aftershock, and the inappropriate location settings of the receiver fault. These issues will be all modified in the revised manuscript by making new inversion based on curved fault and recalculating CFS.

2) In Figure 4c, below the fault plane, the most reliable positive feature is the off-fault lobe of Coulomb stress, which is located at a distance of about 200 km. It is due to tensile stress changes caused by the main rupture (the so-called antithetic lobe).However few of the aftershocks reported in Figure 4b seem to locate there.

**Answer:**

The off-fault lobe of Coulomb stress with positive value appears in the deep near the trench (Figure 4c), We think there may be several factors that can lead to this phenomenon: for example, aseismic slip and low initial stress accumulation in the area. In our revised manuscript, we will give a more reasonable explanation based on new CFS calculation results and relevant referencs.

Lines 273-275 Sentence to be rewritten for clarity.

**Answer:**

We will carefully revise these sentences in the revised manuscript.